# The Value of Tucatinib in Metastatic HER2-Positive Breast Cancer Patients: An Italian Cost-Effectiveness Analysis

**DOI:** 10.3390/cancers15041175

**Published:** 2023-02-12

**Authors:** Ippazio Cosimo Antonazzo, Paolo Angelo Cortesi, Gerardo Miceli Sopo, Giampiero Mazzaglia, Pierfranco Conte, Lorenzo Giovanni Mantovani

**Affiliations:** 1Research Centre on Public Health, University of Milan-Bicocca, 20900 Monza, Italy; 2UOC Farmacia Ospedaliera OP—Continuità Ospedale Territorio e Distribuzione Diretta, ASL Roma 2, Roma Ospedaliera OP—Continuità Ospedale Territorio e Distribuzione, 00157 Ariccia, Italy; 3San Camillo Hospital, IRCCS, Venezia Lido, 30126 Venice, Italy; 4IRCCS, Istituto Auxologico Italiano, 20149 Milan, Italy

**Keywords:** breast cancer, tucatinib, TDM-1, cost-effectiveness analysis, Italy

## Abstract

**Simple Summary:**

Tucatinib is recommended by different guidelines as a third line of treatment in HER2^+^ breast cancer. Although it is an effective treatment, different studies have highlighted that the high price of the drug makes it a not cost-effective treatment option. Therefore, this study aimed to assess the appropriate cost of tucatinib and its cost-effectiveness when used in combination with trastuzumab and capecitabine (TXC) compared with TDM-1 from the perspective of the Italian National Healthcare System. Results suggested that TCX is cost-effective at a willingness to pay (WTP) of 60,000 EUR, using a tucatinib cost of 4828.44 EUR per cycle, in contrast to 4090.60 EUR considering a WTP of 20,000 EUR. Our findings can be used by stakeholders to guarantee the affordability of this drug in the Italian setting and can be used by other European and non-European countries as threshold to establish the appropriate price for the drug.

**Abstract:**

Background: This study was aimed at estimating the appropriate price of tucatinib plus trastuzumab and capecitabine (TXC), as third-line treatment, in HER2^+^ breast cancer (BC) patients from the Italian National Health System (NHS) perspective. Methods: A partitioned survival model with three mutually exclusive health states (i.e., progression-free survival (PFS), progressive disease (PD), and death) was used to estimate the price of tucatinib vs trastuzumab emtansine (TDM-1), considering a willingness to pay (WTP) of 60,000 EUR. Data from the HER2CLIMB trial, the Italian population, and the literature were used as input. The model also estimated the total costs and the life-years (LY) of TXC and TDM1. Deterministic and probabilistic (PSA) sensitivity analyses were conducted to evaluate the robustness of the model. Results: In the base case scenario, the appropriate price of tucatinib was 4828.44 EUR per cycle. The TXC resulted in +0.28 LYs and +16,628 EUR compared with TDM-1. Results were mainly sensitive to therapy intensity variation. In PSA analysis, TXC resulted cost-effective in 53% of the simulations. Assuming a WTP ranging 20,000–80,000 EUR, the tucatinib price ranged from 4090.60 to 5197.41 EUR. Conclusions: This study estimated the appropriate price for tucatinib according to different WTP in order to help healthcare decision makers to better understand the treatment value.

## 1. Introduction

Breast cancer (BC) remains one of the leading causes of morbidity and mortality among women worldwide [1]. In 2020, the BC was the fifth leading cause of cancer mortality with 685,000 deaths worldwide [2]. According to the WHO estimate, BC accounts for one in four cancer cases and one in six cancer deaths [2]. In Italy, BC was responsible for 55,000 new diagnoses, over 834,000 prevalent cases, and about 12,500 deaths due to cancer among women in 2021 [3]. In BC, the molecular subtype might influence the therapeutic approach and the expected clinical outcomes. Human epidermal growth factor receptor 2 (HER2), an oncogene for tumorigenesis, is overexpressed in 15–20% of invasive BC; before the availability of anti-HER2 drugs, HER2-positive (HER2^+^) BC was associated with an increased risk of visceral metastasis and worse outcomes such as shorted progression-free survival and overall survival [4,5,6,7,8]. In the last decades, several anti-HER2 treatments have been approved as first- and second-line treatments which have been associated with significantly improvements in the prognosis of patients with advanced HER2^+^ disease. More recently, the therapeutic armamentarium has been increased, and new effective therapies are available as treatment options for patients with HER2^+^ BC who had no benefit from previous therapies. According to ESMO guidelines, several factors such as the type of prior second-line therapy, patient characteristics, and benefit–risk profile of drugs must be considered when choosing the best option for these patients. Specifically, tucatinib plus trastuzumab and capecitabine (TXC) and trastuzumab emtansine (TDM-1) are the two treatment options recommended as third-line therapies in HER2^+^ BC patients with two failed treatment lines [9]. 

Tucatinib is a small orally bioavailable drug which belongs to the tyrosine kinase inhibitor (TKI) class, which selectively targets HER2 and can go through the blood–brain barrier [10]. Conversely to other TKIs, the high selectivity of this drug for HER2 and lower selectivity for EGFR results in fewer side-effects related to EGFR inhibition [10]. The benefit–safety profile of tucatinib was investigated in the pivotal phase II HER2CLIMB trial (clinicalTrials.gov number: NCT02614794) [11]. Specifically, in the trial, tucatinib was compared with placebo, each in combination with trastuzumab and capecitabine, for HER2^+^ metastatic BC after progression on trastuzumab, pertuzumab, and TDM-1 [11]. Findings from the trial highlighted that, at 1 year of follow-up, the risk of progression or death was 46% lower in the tucatinib plus trastuzumab and capecitabine group compared with placebo (Hazard Ratio [HR]: 0.54; 95% CI: 0.42–0.71). Similarly, at 2 years, the risk of death was 34% lower in the tucatinib combination than in the placebo combination (HR: 0.66; 95% CI: 0.50–0.88) [11]. In addition, in the extension phase of the HER2CLIMB trial, treatment with the tucatinib combination continued to demonstrate a significant survival benefit during 15.6 months of additional follow-up [12]. 

Despite showing excellent efficacy, the price of tucatinib undermines its potential benefits from a value standpoint, whereby pharmacoeconomic evaluations are needed to suggest an acceptable tucatinib price on the basis of its cost-effectiveness profile. For this reason, the present study aimed to estimate a possible appropriate price range of tucatinib in HER2^+^ BC patients with two failed treatment lines from the Italian National Health System (NHS) perspective. 

## 2. Materials and Methods

### 2.1. Study Setting and Overview of the Model

A partitioned survival decisional analytical model based on three mutually exclusive health states, progression-free survival (PFS), progressive disease (PD), and death (Figure 1) [13,14,15], was developed to estimate the cost and effectiveness associated to tucatinib-combination compared to TDM-1. The model was developed in Microsoft Excel, following the ISPOR guidelines [16], from the Italian NHS point of view, considering a 10 year time horizon and a discounted annual rate of 3.0%. The model cycle length was 1 week with no half-cycle correction as suggested by Nemeth and colleagues [17]. In the model, patients’ overall survival (OS) and PFS were determined on the basis of the results of a network meta-analysis. In each model cycle, patients started in the PFS under one of the studied treatment strategies, transitioning to PD or death, or remaining in PFS; once in PD, they could remain as PD, be treated with other anticancer treatments, or transition to death (Figure 1 and Figure A1 in Appendix A). 

In the model, two treatment strategies were compared. In strategy 1, called TXC, patients received as treatment tucatinib (300 mg twice per day) + trastuzumab (8 mg/kg the first day of a 21 day cycle, followed by 6 mg/kg for the remainder of the cycle and the subsequent cycles of treatment) + capecitabine (1000 mg/m^2^ twice per day from day 1 to day 14 per each cycle). In strategy 2, called TDM-1, the patients received as treatment trastuzumab emtansine (3.6 mg/kg every 21 days). Although trastuzumab deruxtecan and neratinib are recommend by guidelines as possible therapeutic options for HER2^+^ BC patients with two failed therapies [9], in Italy, these treatments are still not reimbursed. For this reason, it was not possible to included them as possible therapeutic options in the model. 

The main model outcome was the price of tucatinib using a willingness to pay threshold of 60,000 EUR per LY gained as the base case. To define this price of the tucatinib, the model estimates the overall costs and life-years (LYs) for both the tucatinib combination and TDM-1, as well as the related incremental cost-effectiveness ratio (ICER).

### 2.2. Patients’ Population

The target population of this model was consistent with the HER2CLIMB trial, representing the Italian population with breast cancer [11,18]. Specifically, the model simulated a cohort aged 54 (0.44) years (mean, SE), with HER2^+^ metastatic or recurrent BC who had no benefit from two previous lines of treatments [11] (Table 1). As reported in Table 1, the simulated cohort also had the following characteristics: a body weight of 69.5 kg, with a body surface of 1.8 m^2^ [11], in line with Italian BC population data [18].

### 2.3. Clinical Data Inputs

The OS and PFS probabilities were estimated on the basis of a published network meta-analysis (NMA) [19]. This NMA estimated the hazard rates (HRs) for progression-free survival (PFS) and overall survival (OS) associated with the tucatinib combination and to TDM-1. The HRs used were estimated with a fixed-effects Bayesian model, which was found to be the best model to apply to the available data [19,27]. Estimated HRs were applied to PFS and OS curves of the most studied treatment strategy (i.e., lapatinib plus capecitabine) used, as suggested by NICE guidelines, as reference treatments in the NMA [19,27,28,29,30]. These data were then used to estimate the distribution of patients across the different health states (PFS, PD, and death) during the simulation in both studied treatments. 

The safety data included in the model were derived from the trials of the two treatment strategies [11,22].

### 2.4. Cost Input

The cost of each treatment strategy was assessed from the perspective of the Italian NHS; therefore, only the direct costs were considered in the model. In particular, these encompassed drug costs, drug administration costs, management of adverse events, and disease management across different states and death. 

The treatment costs were estimated on the basis of the ex-factory price for the Italian NHS. This cost was combined with the dose and treatment duration. Treatment duration was derived by constructing the time to treatment discontinuation (TTD) curve on the data from the HER2CLIMB trial for the tucatinib combination using a flexible Weibull (two knots) distribution [11], while, for TDM-1, the curve was constructed using the median treatment time reported in the trial and applying an exponential distribution [22]. Doses were adjusted for relative dose intensity reported in the trials, to estimate the real drug consumptions of the two treatment approaches in the model [11,22]. 

For intravenous drugs (i.e., trastuzumab and TDM-1), the administration costs were based on the national ambulatory tariff [23]. The model simulated the use of loperamide (6 mg per day) for 21.63 days and 5.80 days as supportive therapy, as reported in the HER2CLIMB trial, to manage the most common adverse event (i.e., diarrhea) [11]. The loperamide treatment costs was based on the ex-factory price for the Italian NHS. 

In the tucatinib combination and TDM-1 group, the patients who transitioned to PD status were considered treated with other antineoplastic treatments such as lapatinib, trastuzumab, pertuzumab, and TDM-1. The distribution of patients across the different products was based on HER2CLIMB trial data [11]. For the aforementioned treatments, data from relevant clinical trials were used to estimate the dose and the duration of each treatment option. In particular, the NALA data were used to estimate the parameters for lapatinib [21], along with the HER2CLIMB trial for trastuzumab [11], PHEREXA study for pertuzumab [20], and EMILIA trial for TDM-1 [22]. The costs associated with these treatments were estimated on the basis of the ex-factory price for the Italian NHS. 

The costs associated with severe adverse event (SAE grade ≥ 3) management, which occurred in ≥2% of treated individuals, were based on the Italian tariff for hospitalization (i.e., day hospital and ordinary hospitalization). 

Other disease management costs associated with PFS and PD were applied. In particular, the types of healthcare resources used by breast cancer patients were based on data retrieved from an Italian survey [24]. Lastly, costs were estimated by multiplying the number of healthcare resources used by patients (i.e., medical visits and monitoring visit/exams) with the cost of each resource as reported in the Italian tariff [23,24]. Data from an Italian study were used to estimate the costs associated with the transition of patients from PD to death [25].

### 2.5. Statistical Analysis

For the base case scenario, the appropriate cost for tucatinib was assessed by considering a WTP threshold of 60,000 EUR. Then, this price was used to perform a cost-effectiveness analysis to assess the costs (i.e., direct costs) and the effectiveness (i.e., LYs) of the treatment with the tucatinib combination and TDM-1. The results were expressed as the incremental cost-effectiveness ratio (ICER) expressed as EUR per life-year (LY) gained. 

In addition, a series of deterministic sensitivity analyses were conducted to explore the impact of uncertainty in our assumption on treatment efficacy, utilities, and cost. Specifically, we conducted a one-way sensitivity analysis to iteratively replace individuals in the model input using their 95% confidence intervals or alternatively high and low range values (calculated using the mean and standard error) and re-estimated model results, while holding other inputs constant. Furthermore, we ranked the resulting set of sensitivity analyses by the absolute magnitude of deviation from the base case to assess which input parameters mostly affected the results. In addition, we performed alterative scenario analysis, investigating the appropriate cost of tucatinib according to the range of the WTP threshold (0–80,000 per LY gained). Then, we performed a 1000 Monte Carlo simulation to conduct probabilistic sensitivity analyses for each explored alternative scenario of the tucatinib price. The results of these analyses were expressed as cost-effectiveness acceptability curves. 

Lastly, we performed a second alternative scenario analysis to assess the impact of TDM-1 price reduction on the appropriate tucatinib cost estimated at a threshold of 60,000 EUR per LY gained. This analysis provided data to understand the appropriate price of tucatinib as a function of the possible lower price of TDM-1 on the market. 

## 3. Results

Base Case Results

Considering a WTP of 60,000 EUR, the appropriate price of tucatinib estimated by the model was 4828.44 EUR per treatment cycle (21 days). Results of the model projected that the patients treated with TXC yielded 3.11 LYs, which was 0.28 LYs more than patients received TDM-1. The use of TXC also resulted in an overall cost of 125,710 EUR, which was 16,628 EUR more than patients treated with TDM-1 (Table 2). 

The tornado diagram from the base case scenario created with the one-way sensitivity analysis in shown in Figure 2. The most influential parameters resulting from the analysis were the TDM-1 intensity dose, TXC intensity dose, and weight (kg) of treated patients. The remaining parameters weakly affected the estimated ICER and, consequently, the price of tucatinib.

In the alternative scenario analysis, a range of different WTP thresholds were tested to estimate a possible range of tucatinib costs as a function of willingness to pay. Assuming a WTP of 80,000 EUR per LY gained, the tucatinib price per treatment raised to 5197.41 EUR per cycle; assuming a WTP of 40,000 and 20,000 EUR, the tucatinib cost decreased to 4459.52 and 4090.60 EUR per cycle. The cost of tucatinib decreased to 3721.65 EUR per cycle if we assumed a WTP of 0 (no additional cost associated with the tucatinib combination compared to TDM-1 (Figure A2 in Appendix A).

Results of the probabilistic sensitivity analysis are reported in Figure 3. In the base case scenario, considering a WTP of 60,000 EUR per LYs gained, the tucatinib combination resulted cost-effective compared with TDM-1 in 53% of the simulation (Figure 3). The estimated probability to be cost-effective considering a WTP of 60,000 EUR increased in the alternative scenarios in which the tucatinib price was lower than reported in the base case scenario. In particular, the probability for tucatinib to be cost-effective compared with TDM-1 increased from 58% to 75% when considering tucatinib costs of 4459.52 EUR (ICER of 40,000 EUR per LYs) and 3721.65 EUR (ICER of 0 EUR per LYs gained) per cycle, respectively (Figure 3). 

In the second alternative scenario analysis, a range of different TDM-1 percentage price reductions were tested to estimate a possible range of tucatinib costs according to different TDM-1 prices. Assuming a TDM-1 price reduction of 20%, the tucatinib costs decreased to 3851.00 EUR per cycle; assuming a TDM-1 price reduction of 40%, the appropriate tucatinib cost decreased to 2872.63 EUR per cycle (Figure A3 in Appendix A).

## 4. Discussion

To our knowledge, our study is the first economic analysis of the tucatinib combination in European countries. Specifically, our study attempted to assess the appropriate price range of tucatinib as a potential third-line treatment in patients with HER2^+^ breast cancer who had no benefit from the previous therapies. In the base case scenario, considering a WTP of 60,000 EUR per LY gained, tucatinib should cost at maximum 4828.44 EUR per treatment cycle. This price should be lowered to 3721.65 EUR per cycle when a parity price with the comparator is warranted by the NHS. Considering the one-way sensitivity and alternative scenario analyses, the model was robust, and the variation of the ICERs was mainly ascribed to the costs of tucatinib and TDM-1. The impact of the cost of tucatinib on the ICER is consistent with what has already been observed in previous cost-effectiveness analyses for different innovative drugs for the treatment of advanced cancer [31,32,33,34,35]. 

Our findings on OWSA are in line with a previous study on tucatinib as treatment in HER2^+^ BC conducted by Wu and colleagues [36]. In the aforementioned study, the authors highlighted that the cost of tucatinib was the main driver for the cost-effectiveness of therapy with tucatinib, suggesting that this drug was not a cost-effective treatment for the USA and China setting by considering its marketed price [36]. In the aforementioned study, the price of tucatinib was the parameter that, in the OWSA, resulted in a tremendous variation of the ICER. Specifically, this ranged from 77,189 to 105,715 USD for China and from 527,420 to 885,926 USD for the USA, which were, in both cases, higher than the WTP threshold set in each country [36]. In another study, the cost-effectiveness of tucatinib in HER2^+^ BC patients with brain metastasis was explored. Similarly, to the previous study, the authors reported an ICER of over 400,000 USD per QALY, which was higher than the WTP threshold of 200,000 USD per QALY in the country; therefore, the drug resulted as a non-cost-effective treatment option [37] Consequently, the authors highlighted the importance of reducing the price of tucatinib to approximately 70% at the given WTP to increase the probability of the drug being cost-effective beyond 50% [37]. It should be noted that a comparison between our findings and those reported might be misleading due to difference in study aims and outcomes. In fact, previous studies assessed the cost-effectiveness of tucatinib given a specific price. On the contrary, in this study, we estimated the most appropriate price for the treatment by considering different WTPs according to the national threshold. 

The strengths of this study are worth highlighting. Firstly, the lack of cost-effectiveness analyses for tucatinib in a European setting provides an insufficient reference for decision makers. Our study evaluated the potential price of tucatinib according to the beneficial effect compared with the standard of care as a third line of treatment in patients with HER2^+^ BC, providing for the first time a range of prices for the drug in order to be cost-effective in the country. Secondly, our careful delineation and consideration of costs (i.e., the cost for supportive therapy, management of adverse events, correction of utility value of metastasis, and adverse event occurrence) and the impact of treatment, as well as adverse events, on QoL contribute to the reliability of our findings. Lastly, we carried out a series of scenario analyses to dissect the price of tucatinib under different WTP and TDM-1 prices to make the study more informative and the findings more convincing and applicable to the country setting. In this regard, it should be noted that different factors might impact the price and reimbursement by the National Healthcare system of a drug. Our findings can be used by stakeholders during the negotiation phase in Italy between the Italian Medicine Agency (AIFA) and the pharmaceutical companies. Additionally, an established cost-effectiveness threshold for drug evaluation does not exist in Italy [38]; therefore, we estimated the price of tucatinib by considering different WTPs in order to provide the best possible evidence on the potential price for the drug. 

As also observed for other pharmacoeconomic evaluations, this study was subject to limitations. Firstly, we made assumptions in fully specifying the model to translate the complex medical decision process and treatment pathway in the model structure used for the analysis. For example, we assumed that the effectiveness of the studied treatments in the real-world setting was similar to the efficacy shown by treatment in the randomized controlled trials. Secondly, in the model we included only direct medical costs. Despite costs in patients with advanced cancer being mostly direct costs, patients may also have indirect costs that should be considered in future analysis. However, in Italy, we can only capture direct costs because they are fully reimbursed by the NHS. Lastly, in the model, we assumed that all patients received the best supportive care after progression. Although, in Italy, most medications are reimbursed by NHS, this assumption might not be completely consistent with the real clinical practice.

## 5. Conclusions

From the perspective of the Italian NHS, to guarantee the cost-effectiveness of the treatment with TXC, the cost of tucatinib should be 4828.44 EUR per cycle considering a WTP of 60,000 EUR per LY gained. The most important driver for the cost-effectiveness of tucatinib was the price of tucatinib and TDM-1. Considering the availability of a broad range of costly drugs, health policymakers should consider this type of analysis in order to guarantee the affordability of new drugs for the healthcare system. Specific subpopulation data for the compared treatments are needed to assess the cost-effectiveness of the study drug across different subgroups of BC patients, providing additional information for healthcare decision makers. Lastly, similar studies should be performed in other European and non-European countries in order to confirm our results. 

## Figures and Tables

**Figure 1 cancers-15-01175-f001:**
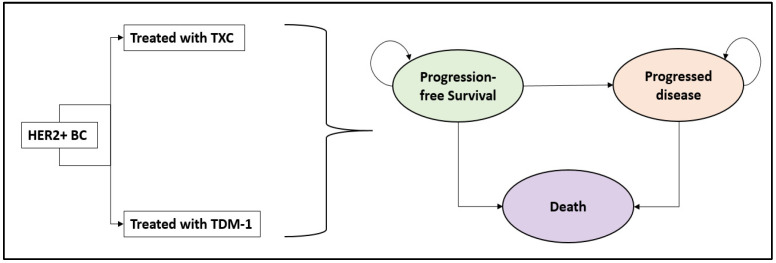
The partitioned survival model. BC: Breast cancer TXC: tucatinib plus trastuzumab and capecitabine; TDM-1: trastuzumab emtansine.

**Figure 2 cancers-15-01175-f002:**
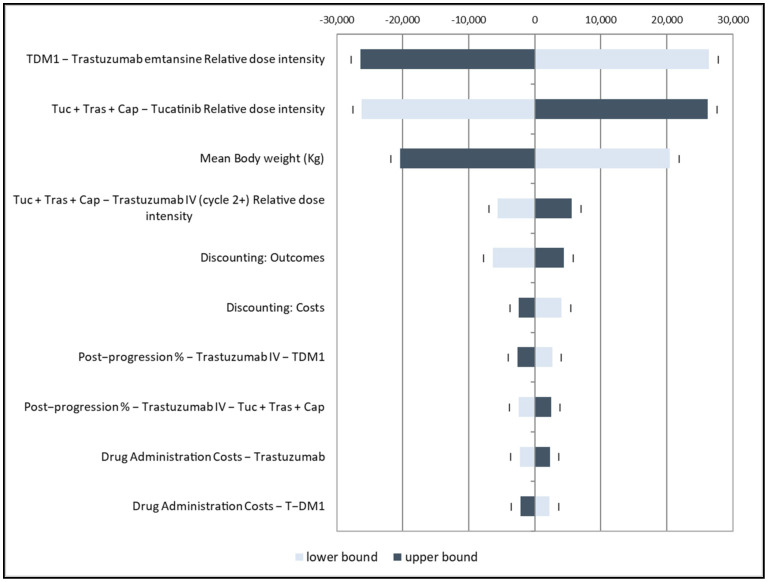
Tornado diagram.

**Figure 3 cancers-15-01175-f003:**
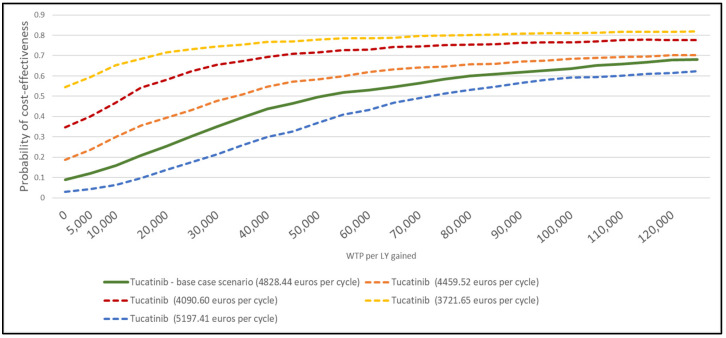
Acceptable curves of cost-effective probability of tucatinib, trastuzumab, and capecitabine at different prices from the Italian NHS perspective.

**Table 1 cancers-15-01175-t001:** Model input.

Parameters	Value	Range of Variation (Confidence Interval or Standard Error)	Distribution	Source
**Clinical input**				
Age (mean)	54 years	0.44	Normal	[11]
Body surface (mean)	1.8 m^2^	0.18	Normal	[11]
Weight (Kg)	69.5 kg	6.95	Normal	[11]
**Treatment efficacy vs. lapatinib + capecitabine (PFS)**				
Tucatinib + trastuzumab + capecitabine	0.56	0.50–0.61	Log-normal	[19]
TDM-1	0.65	0.59–0.72	Log-normal	[19]
**Efficacy of treatments vs. lapatinib + capecitabine (Survival)**				
Tucatinib + trastuzumab + capecitabine	0.60	0.54–0.66	Log-normal	[19]
TDM-1	0.70	0.63–0.77	Log-normal	[19]
**Treatment duration**				
Tucatinib + trastuzumab + capecitabine	Flexible Weibull—2 knots (mean duration in months: 12.3)	-	Normal multivariate	[11]
TDM-1	Exponential (mean duration in months: 11.11)	-	Exponential	[18]
**Treatment duration in the post-progression phase (months)**				
Trastuzumab	5.70	0.31	Normal	[11]
Pertuzumab	10.35	1.03	Normal	[20]
Lapatinib	4.4	0.44	Normal	[21]
T-DM1	9.60	0.96	Normal	[22]
**Duration of antidiarrheal treatment (loperamide) in days**				
Tucatinib + trastuzumab + capecitabine	21.63	2.16	Normal	[11]
T-DM1	5.80	0.58	Normal	[11]
**Dose intensity**				
* tucatinib + trastuzumab + capecitabine *				
Tucatinib	88.5%	0.01	Beta	[11]
Capecitabine	73.9%	0.01	Beta	[11]
Trastuzumab (cycle 1)	100%	-	Fixed	[11]
Trastuzumab (cycle 2+)	73.9%	0.01	Beta	[11]
* T-DM1 *				
T-DM1	97.2%	0.01	Beta	[22]
**Adverse events, grade ≥ 3**				
* tucatinib + trastuzumab + capecitabine *				
Hand–foot syndrome	13.1%	α = 53, β = 351	Beta	[11]
Diarrhea	12.9%	α = 52, β = 352	Beta	[11]
Alanine aminotransferase increased	5.4%	α = 22, β = 382	Beta	[11]
Fatigue	4.7%	α = 19, β = 385	Beta	[11]
Aspartate aminotransferase increased	4.5%	α = 18, β = 386	Beta	[11]
Anemia	3.7%	α = 15, β = 389	Beta	[11]
Nausea	3.7%	α = 15, β = 389	Beta	[11]
Neutropenia	0.0%	α = 0, β = 404	Beta	[11]
Vomiting	3.0%	α = 12, β = 392	Beta	[11]
Hypokalemia	0.0%	α = 0, β = 404	Beta	[11]
Inflammation of mucous membrane	0.0%	α = 0, β = 404	Beta	[11]
Thrombocytopenia	0.0%	α = 0, β = 404	Beta	[11]
Stomatitis	2.5%	α = 10, β = 394	Beta	[11]
* TDM-1 *				
Hand–foot syndrome	0.00	α = 0, β = 490	Beta	[22]
Diarrhea	0.02	α = 8, β = 482	Beta	[22]
Alanine aminotransferase increased	0.03	α = 14, β = 476	Beta	[22]
Fatigue	0.02	α = 12, β = 478	Beta	[22]
Aspartate aminotransferase increased	0.04	α = 21, β = 469	Beta	[22]
Anemia	0.03	α = 13, β = 477	Beta	[22]
Nausea	0.01	α = 4, β = 486	Beta	[22]
Neutropenia	0.02	α = 10, β = 480	Beta	[22]
Vomiting	0.01	α = 4, β = 486	Beta	[22]
Hypokalemia	0.02	α = 11, β = 479	Beta	[22]
Inflammation of mucous membrane	0.00	α = 1, β = 489	Beta	[22]
Thrombocytopenia	0.13	α = 63, β = 427	Beta	[22]
Stomatitis	0.00	α = 0, β = 490	Beta	[22]
**Monthly costs per health status (EUR)**				
Progression-free	98.17	9.82	Gamma	[23,24]
Progression	98.17	9.82	Gamma	
Death	14,316.00	1431.60	Gamma	[25]
**Costs for each adverse event of grade ≥ 3 (EUR)**				
Hand–foot syndrome	728.00	72.8	Gamma	[23]
Diarrhea	238.00	23.8	Gamma	
Alanine aminotransferase increased	236.00	23.6	Gamma	
Fatigue	209.00	20.9	Gamma	
Aspartate aminotransferase increased	236.00	23.6	Gamma	
Anemia	1676.00	167.6	Gamma	
Nausea	238.00	23.8	Gamma	
Neutropenia	1993.00	199.3	Gamma	
Vomiting	238.00	23.8	Gamma	
Hypokalemia	216.00	21.6	Gamma	
Inflammation of mucous membrane	222.00	22.2	Gamma	
Thrombocytopenia	2748.00	274.8	Gamma	
Stomatitis	269.00	26.9	Gamma	
**Costs of therapies (EUR)**				
Capecitabine (500 mg × 120)	113.67	Fixed		[26]
Trastuzumab (150 mg)	1294.63	Fixed		[26]
TDM-1 (100 mg)	1837.32	Fixed		[26]
**Costs of drugs used during post-progression phase (EUR)**				
Lapatinib (250 mg × 84)	1326.67	Fixed		[26]
Pertuzumab (420 mg)	2885.91	Fixed		[26]
**Cost of supportive therapy (EUR)**				
Loperamide (2 mg × 30)	2.59	Fix		[26]
**Treatment during post-progression phase: treatment subsequent to tucatinib + trastuzumab + capecitabine**				
Trastuzumab	51.0%	α = 149, β = 143	Beta	[11]
Lapatinib	13.4%	α = 39, β = 253	Beta	[11]
Pertuzumab	4.0%	α = 12, β = 280	Beta	[11]
TDM-1	1.8%	α = 5, β = 287	Beta	[11]
**Administration costs (EUR)**				
Tucatinib	0	-		Assumption
Capecitabine	0	-		Assumption
Trastuzumab	371.00	37.10	Gamma	[23]
Pertuzumab	371.00	37.10	Gamma	
T-DM1	371.00	37.10	Gamma	
Lapatinib	0	-		Assumption

**Table 2 cancers-15-01175-t002:** Cost-effectiveness analysis results for the base case scenario.

	Costs (EUR)	Delta Costs	LYs	Delta LYs	ICER (EUR Per LY Gained)
TDM-1	109,082		2.83		
Tucatinib + trastuzumab + capecitabine	125,710	16,628	3.11	0.28	60,000

## Data Availability

The data can be shared up on request.

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
