# Peer review of "The Value of Tucatinib in Metastatic HER2-Positive Breast Cancer Patients: An Italian Cost-Effectiveness Analysis"

_cancers, 2023, doi:10.3390/cancers15041175_

Round 1

Reviewer 1 Report

Nicer paper. It should be publlished.

Author Response

We thank the reviewer for the comment.

Reviewer 2 Report

The article proposed by the authors has many theoretical aspects, not always applicable to the Italian healthcare reality. An Italian cost-effectiveness analysis cannot be done without also considering the Italian public health situation and the many variables of the real health world and of the Italian socio-economic reality. In Italy we are heavily penalized because we have 20 different regional health systems, which do not communicate with each other, behave completely independently, and represent 20 different variables that heavily affect any health topic we want to discuss in a global.

The prices requested by pharmaceutical companies for the latest generation of drugs are reaching worrying levels. An innovative anticancer therapy can easily cost 100,000 euros per patient per year. For some genetic diseases, the price can be double or triple. These are costs that certainly weigh on the state coffers, seriously endangering the sustainability of health services.

It is a trend that afflicts all countries, more or less rich.

It may seem only a clinical and healthcare problem, which does not directly concern citizens, since the State and the Public Health Service should be able to find the funds to pay for innovative treatments. In reality, the problem is social. When a drug is too expensive, citizens are sometimes forced to wait months before being able to receive it, i.e. the time necessary to complete the lengthy negotiations between the manufacturer and the Italian drug agency, seeking an agreement on the reimbursement price. But, even when a medicine is included among the reimbursed medicines, the expenditure weighs on the insufficient resources of the state budget allocated to public health, with the effect of eroding the offer of health services and benefits that are just as useful, if not more, in areas that are often chronically lacking or in difficulty. As an example, we can consider the medical assistance to the disabled or, more simply, the containment of waiting lists for specialist visits and diagnostic tests, which in fact afflict all twenty regions. The pandemic has definitively brought down this part of public assistance, everywhere in Italy.

ISTAT says that in Italy, in 2017, 20.2 billion euros were spent on pharmaceutical expenditure supported by the NHS and overall public health cost the State 113 billion, a figure already undersized to real needs, with a 50% increase in spending on hospital drugs from 2011 to 2017. Growth largely because of the high incidence of the cost of new drugs. The trend continues even in the period of the pandemic but with much less reliable data. In Italy, there is no single, national public repository where all the medical information of all the patients managed by the public health are archived. In this way, the right to care becomes increasingly uncertain because these costs prevent other treatments and, above all, prevention. For the very expensive new generation drugs, the standard acquisition negotiations (AIFA) are rigorously covered by confidentiality and lengthy. The so-called "ex-factory" price is not what the National Health Service actually pays, but it is the basis from which the pharmaceutical company starts for negotiation.

Therapeutic advance can be provided in three ways: better efficacy; fewer adverse effects; better convenience to patients. Some time ago we had the case of Zolgensma (Novartis), whose price per dose was set at around $2.1 million. Unfortunately, it is one of only two therapies that currently exist to treat spinal muscular atrophy (SMA). Even anti-cancer therapies such as the so-called "Car-T" are yet to come. It has already happened for many drugs which, because of the astronomical price, were initially rationed, favoring patients in more serious conditions. It was a socially and legally dangerous precedent, because the principles of universality and equity of access to treatment on which our national health service is based were betrayed. No one can be denied the right to health. If there are diseases that are more serious than others, there are no patients who are less deserving of treatment. But the new does not always mean innovation. Out of the drugs allowed in the last 10 years, only 1% are a real therapeutic innovation, and for those with innovative potential we are around 7%. Even if the pharmaceutical companies say that the high cost is the only way to support the costs of innovation, it must be said that more and more often the pharmaceutical multinationals, rather than carrying out research in their own laboratories, buy the patents of products developed elsewhere, in smaller companies, perhaps linked to public universities or which have received public funding.

Another controversial point is the origin of the cancer data in Italy. The authors cannot claim that the data is from the Italian National Health System (NHS), because such data does not exist. The Cancer Registries are private structures engaged in an associative form in the collection of information on cancer patients living in a given territory. In Italy there is a formal coverage of 70% but in this 70% not all registers are already functional. There is no single national register. Regardless of the certain merit of these associations, the overall error they carry in their data and information is of the order of 40%, too high for any meaningful statistics on cancer and different cancer. In Italy, the fundamental reason for the high uncertainty and low reliability of public and private assistance data is because the general process of digitizing data and related information is in fact still non-existent after 10 years of attempts. If there is not a single national public information system suitable for the times, organized, interoperable, integrated, containing the health documents of all Italian citizens, suitably structured in HL7-FIHR, with the international codes, with the semantic ontologies, how can the authors confidently state that breast cancer data and information is real and statistically reportable to the whole country? Any other non-public source is not certified and there is no certainty that the data represents the entire national territory. All this should be based on a fundamental object, called the Electronic Health Record (FSE), which does not yet exist. Even the most advanced regions in the sector (Lombardy and Emilia and Romagna) alone cannot say anything.

The authors, in their analysis which uses mainly statistical mathematical models, should declare all the errors that their analyzes involve, from which reliable and public sources they have extracted their data and their beliefs. In its present form, the article is an appreciable theoretical presentation of a general analysis model, but certainly not relevant to Italy as a whole.

Figure 3 is difficult to read because all characters are tiny.

The authors should also specify if there are patients older than 75-80 years in their sampling, for whom tucatinib has not been studied.

The acronym RCT (line 281) should be explicated (I think is for Randomized Controlled Trials)

Line 277 and following.

The authors state that the study is subject to limitations. They make assumptions that, for the reasons explained above, are difficult to make, because efficacy and effectiveness are highly variable. In fact, we have 20 different health systems that follow different and sometimes inappropriate rules. So, on average, far from clinical practice in the real world. However, the efficacy can be showed in an explanatory, i.e., a randomized controlled trial (RCT) completed under ideal study conditions and with affordable data at high veracity (veracity, i.e. the accuracy of the data, is one of the five fundamental properties of Big Data and in particular of Big Biomedical Data on which clinical statistics on patients are based. Data veracity should be below 1% to have significant data. In short, accuracy of big data, it's not just the quality of the data itself but how trustworthy the data source, type, and processing of it is. How did the authors get this certainty?). Effectiveness can be showed in an observational, i.e., a pragmatic controlled trial (PCT) completed under real-world conditions. I suggest setting the trial in Calabria or Sicily.

The current version of the manuscript is not suitable for the publication.

Author Response

Reviewer 2

We thank the reviewer for the comments on our manuscript which gave us the possibility to improve it. In the new version of the manuscript we carefully considered all raised points. 

Our point-by-point responses are provided below. 

The article proposed by the authors has many theoretical aspects, not always applicable to the Italian healthcare reality. An Italian cost-effectiveness analysis cannot be done without also considering the Italian public health situation and the many variables of the real health world and of the Italian socio-economic reality. In Italy we are heavily penalized because we have 20 different regional health systems, which do not communicate with each other, behave completely independently, and represent 20 different variables that heavily affect any health topic we want to discuss in a global.

The prices requested by pharmaceutical companies for the latest generation of drugs are reaching worrying levels. An innovative anticancer therapy can easily cost 100,000 euros per patient per year. For some genetic diseases, the price can be double or triple. These are costs that certainly weigh on the state coffers, seriously endangering the sustainability of health services.

It is a trend that afflicts all countries, more or less rich.

It may seem only a clinical and healthcare problem, which does not directly concern citizens, since the State and the Public Health Service should be able to find the funds to pay for innovative treatments. In reality, the problem is social. When a drug is too expensive, citizens are sometimes forced to wait months before being able to receive it, i.e. the time necessary to complete the lengthy negotiations between the manufacturer and the Italian drug agency, seeking an agreement on the reimbursement price. But, even when a medicine is included among the reimbursed medicines, the expenditure weighs on the insufficient resources of the state budget allocated to public health, with the effect of eroding the offer of health services and benefits that are just as useful, if not more, in areas that are often chronically lacking or in difficulty. As an example, we can consider the medical assistance to the disabled or, more simply, the containment of waiting lists for specialist visits and diagnostic tests, which in fact afflict all twenty regions. The pandemic has definitively brought down this part of public assistance, everywhere in Italy.

ISTAT says that in Italy, in 2017, 20.2 billion euros were spent on pharmaceutical expenditure supported by the NHS and overall public health cost the State 113 billion, a figure already undersized to real needs, with a 50% increase in spending on hospital drugs from 2011 to 2017. Growth largely because of the high incidence of the cost of new drugs. The trend continues even in the period of the pandemic but with much less reliable data. In Italy, there is no single, national public repository where all the medical information of all the patients managed by the public health are archived. In this way, the right to care becomes increasingly uncertain because these costs prevent other treatments and, above all, prevention. For the very expensive new generation drugs, the standard acquisition negotiations (AIFA) are rigorously covered by confidentiality and lengthy. The so-called "ex-factory" price is not what the National Health Service actually pays, but it is the basis from which the pharmaceutical company starts for negotiation.

Therapeutic advance can be provided in three ways: better efficacy; fewer adverse effects; better convenience to patients. Some time ago we had the case of Zolgensma (Novartis), whose price per dose was set at around $2.1 million. Unfortunately, it is one of only two therapies that currently exist to treat spinal muscular atrophy (SMA). Even anti-cancer therapies such as the so-called "Car-T" are yet to come. It has already happened for many drugs which, because of the astronomical price, were initially rationed, favoring patients in more serious conditions. It was a socially and legally dangerous precedent, because the principles of universality and equity of access to treatment on which our national health service is based were betrayed. No one can be denied the right to health. If there are diseases that are more serious than others, there are no patients who are less deserving of treatment. But the new does not always mean innovation. Out of the drugs allowed in the last 10 years, only 1% are a real therapeutic innovation, and for those with innovative potential we are around 7%. Even if the pharmaceutical companies say that the high cost is the only way to support the costs of innovation, it must be said that more and more often the pharmaceutical multinationals, rather than carrying out research in their own laboratories, buy the patents of products developed elsewhere, in smaller companies, perhaps linked to public universities or which have received public funding.

Another controversial point is the origin of the cancer data in Italy. The authors cannot claim that the data is from the Italian National Health System (NHS), because such data does not exist. The Cancer Registries are private structures engaged in an associative form in the collection of information on cancer patients living in a given territory. In Italy there is a formal coverage of 70% but in this 70% not all registers are already functional. There is no single national register. Regardless of the certain merit of these associations, the overall error they carry in their data and information is of the order of 40%, too high for any meaningful statistics on cancer and different cancer. In Italy, the fundamental reason for the high uncertainty and low reliability of public and private assistance data is because the general process of digitizing data and related information is in fact still non-existent after 10 years of attempts. If there is not a single national public information system suitable for the times, organized, interoperable, integrated, containing the health documents of all Italian citizens, suitably structured in HL7-FIHR, with the international codes, with the semantic ontologies, how can the authors confidently state that breast cancer data and information is real and statistically reportable to the whole country? Any other non-public source is not certified and there is no certainty that the data represents the entire national territory. All this should be based on a fundamental object, called the Electronic Health Record (FSE), which does not yet exist. Even the most advanced regions in the sector (Lombardy and Emilia and Romagna) alone cannot say anything.

The authors, in their analysis which uses mainly statistical mathematical models, should declare all the errors that their analyzes involve, from which reliable and public sources they have extracted their data and their beliefs. In its present form, the article is an appreciable theoretical presentation of a general analysis model, but certainly not relevant to Italy as a whole.

Thank you for the comment. We would like to clarify that this is not an observational study with patients’ data collection from Italy or any other country, instead it is a health-economic evaluation based on decisional analytical model. A simulation study that uses data available in the literature to make forecast on cost and effectiveness of compare treatment on target population. (Caro JJ, Briggs AH, Siebert U, Kuntz KM; ISPOR-SMDM Modeling Good Research Practices Task Force. Modeling good research practices--overview: a report of the ISPOR-SMDM Modeling Good Research Practices Task Force--1. Value Health. 2012 Sep-Oct;15(6):796-803).

Data input for this type of analysis is generally retrieved from the best published evidence available in the literature and adjusted for the country-specific setting if available. In our case, some data on cancer, and costs were retrieved from Italian report on cancer, the Italian NHS tariff and Italian drug costs. For each parameter included in the model, the data source was referenced in table 1. The uncertainty associated to each parameter was assessed in the sensitivity analyses which included the one-way sensitivity analysis and the probabilistic sensitivity analysis as recommended by the International guidelines for health-economic evaluation based on decisional analytical model (Briggs AH, Weinstein MC, Fenwick EA, Karnon J, Sculpher MJ, Paltiel AD; ISPOR-SMDM Modeling Good Research Practices Task Force. Model parameter estimation and uncertainty analysis: a report of the ISPOR-SMDM Modeling Good Research Practices Task Force Working Group-6. Med Decis Making. 2012 Sep-Oct;32(5):722-32).

In the new version of the manuscript we have specify that the model was created following the ISPOR guideline “….The model was developed in Microsoft excel, following the ISPOR guideline [16], and used the Italian NHS point of view, a 10-years’ time horizon and a discounted annual rate of 3.0%.....” and it is based on a decision analytical model approach. [Lines 86-91]

Figure 3 is difficult to read because all characters are tiny.

Thank you for the comment, the table has been updated.  

The authors should also specify if there are patients older than 75-80 years in their sampling, for whom tucatinib has not been studied.

Thank you for the comment, that give use the opportunity to better specify the model input. This is a simulation study based on decision analytical model which includes aggregate data from clinical trials, therefore it is not possible for us to make additional analysis for subgroup population. The model did not include individual patient data and simulate a cohort of patients with an average (SE) age of 54 years (0.44). This aspect has been specified and the follows sentence has been added in the methods section “…Specifically, the model simulated a cohort aged 54 (0.44) years (mean, SE), with HER2+ metastatic or recurrent BC who have no benefit by previous 2 lines of treatments [11]…”. [lines 119-121]

The acronym RCT (line 281) should be explicated (I think is for Randomized Controlled Trials).

Thank you the reviewer for the comment. In the new version of the manuscript we had specify the acronym RCT (Randomized Controlled Trial).  [line 296]

Line 277 and following.

The authors state that the study is subject to limitations. They make assumptions that, for the reasons explained above, are difficult to make, because efficacy and effectiveness are highly variable. In fact, we have 20 different health systems that follow different and sometimes inappropriate rules. So, on average, far from clinical practice in the real world. However, the efficacy can be showed in an explanatory, i.e., a randomized controlled trial (RCT) completed under ideal study conditions and with affordable data at high veracity (veracity, i.e. the accuracy of the data, is one of the five fundamental properties of Big Data and in particular of Big Biomedical Data on which clinical statistics on patients are based. Data veracity should be below 1% to have significant data. In short, accuracy of big data, it's not just the quality of the data itself but how trustworthy the data source, type, and processing of it is. How did the authors get this certainty?). Effectiveness can be showed in an observational, i.e., a pragmatic controlled trial (PCT) completed under real-world conditions. I suggest setting the trial in Calabria or Sicily.

We thank the reviewer for the comment. We agree with the reviewer that in Italy exist 20 different regions, however, the price of a drug and its potential reimbursement is established by the Italian medicine agency (AIFA) for the country. We also agree with the reviewer that unfortunately the access to the health system might be different across the different regions however this is not something that impact on the cost of the drug that is decided by AIFA. As regard observational data for data input into model; although observational studies represent, if available, a suitable source of data, at the moment the best available data on tucatinib efficacy comes from the clinical trials. These data have been used by the European agency and international society (e.g. ESMO) for the treatment approval and for clinical recommendations of its use.

The suggestion of the reviewer to set a trial in Calabria or Sicily where it is possible to suppose that patients might have a different access to healthcare facilities compared with region in the northern Italy, although potentially interesting it is something that is out of scope of our study. However, this might represent a useful input for future studies.

The current version of the manuscript is not suitable for the publication.

We thank the reviewer for the comment and we hope that the new version of the manuscript is suitable for the publication.

Reviewer 3 Report

Pharmacoeconomic evaluation is quite helpful for estimating suitable drug price and provides valuable reference to help health policy makers to understand the treatment value. In this study, the authors devoted to assessing the appropriate price of Tucatinib by using the Italian National Health System (NHS) perspective, because the current price of this drug makes it not a cost-effective treatment option in HER2+ breast cancer, despite of its excellent efficacy. Based on the cost effectiveness profile of the Tucatinib, plus trastuzumab and capecitabine (TXC) treatment, the cost of tucatinib should be 4,828.44 euros and 4,090.60 euros per cycle considering willing to pay (WTP) of 60,000 euros and 20,000 euros, respectively. I have several issues/questions as list below:

1) For the third line therapies in HER2+ BC patients, is there any alternative treatment options instead of TXC and TDM-1?

2) Will the cost-effectiveness of TXC be influenced by the situation in other countries like the USA and the European countries?

3) For HER2+ BC patients in different stages, the cost-effectiveness profile should be variant. The result would be even more helpful if the authors can provide further evaluation regarding the status of different sub-population of the patients. 

Author Response

Reviewer 3

Pharmacoeconomic evaluation is quite helpful for estimating suitable drug price and provides valuable reference to help health policy makers to understand the treatment value. In this study, the authors devoted to assessing the appropriate price of Tucatinib by using the Italian National Health System (NHS) perspective, because the current price of this drug makes it not a cost-effective treatment option in HER2+ breast cancer, despite of its excellent efficacy. Based on the cost effectiveness profile of the Tucatinib, plus trastuzumab and capecitabine (TXC) treatment, the cost of tucatinib should be 4,828.44 euros and 4,090.60 euros per cycle considering willing to pay (WTP) of 60,000 euros and 20,000 euros, respectively. I have several issues/questions as list below:

We thank the reviewer for the comments on our manuscript which gave us the possibility to improve it. In the new version of the manuscript we carefully considered all raised points. 

Our point-by-point responses are provided below. 

  • For the third line therapies in HER2+ BC patients, is there any alternative treatment options instead of TXC and TDM-1?

We thank the reviewer for the comment which give us the possibility to deeper specify the reason beyond our model assumption. Specifically, guideline such as ESMO guideline for HER2+ Breast cancer patients who have failed two lines of treatment recommended the use of TXC, TDM-1, trastuzumab deruxtecan as well as Neratinib. However, in the model, we included only TXC and TDM-1 because trastuzumab deruxtecan and neratinib are not reimbursed by the Italian National Healthcare system. We have better specify this aspect in the methods section by adding the following sentence “Although trastuzumab deruxtecan and neratinib were recommend by guidelines as possible therapeutic option for HER2+ BC patients with two failed therapies [9], in Italy these treatments are still not reimbursed. For this reason, it was not possible to included them as potential therapeutic options in the model…”. [lines 103-107].

  • Will the cost-effectiveness of TXC be influenced by the situation in other countries like the USA and the European countries?

We thank the reviewer for the comment. In Italy, as well as in other European countries, several factors influence the final price and the possibility for its reimbursement by the National Healthcare system. For example, added benefit, expected budget impact, cost-effectiveness findings and published literature play a role in the negotiation process. However, the results of our analysis are mainly related to the cost set for the Italian NHS. The situation in other countries, or the price already set for tucatinib in other countries, could influence the negotiation between the manufacture and the Italian agency. However, this aspect has no impact on our analysis because we aimed to define a price of tucatinib for the Italian context base on efficacy and Italian cost criteria. Our results could be integrated with the situation in other countries to make a decision on the right price for the Italian setting.

Additionally, it should be noted that in Italy an explicit cost-effectiveness threshold does not exist, for this reason we estimated different prices according with different thresholds. In the new version of the manuscript we have better specify this aspect by adding the following sentence “... In this regards it should be noted that different factors might impact on the price and reimbursement by the National Healthcare system of a drug. Our findings, might be used by stakeholders during negotiation phase that in Italy is between the Italian Medicine Agency (AIFA) and the pharmaceutical companies. Additionally, in Italy does not exist an established cost-effectiveness threshold for drug evaluation [38], therefore we estimated the price of tucatinib by considering different WTPs in order to provide as much as possible evidence on the potential price for the drug…”.  [lines 285-291]

3) For HER2+ BC patients in different stages, the cost-effectiveness profile should be variant. The result would be even more helpful if the authors can provide further evaluation regarding the status of different sub-population of the patients. 

Thank you the reviewer for the comment. We agree with the reviewer that subpopulations analyses might be useful for cost-effectiveness analysis. However, this study is based on analysis of all population included in tucatinib trial, with no access of subpopulation data. Further, for subpopulation analysis we need same type of data for the comparators (TDM-1) that at the moment are missing. For this reason, we add the follows sentence in the conclusion of the manuscript “…Specific subpopulation data for the compares treatments are need to assess the cost-effectiveness of the study drug across different subgroups of BC patients, providing additional information useful for healthcare decision makers. Finally, similar studies should be performed in other European and non-European countries in order to confirm or infirm our results…”. [lines 310-315]

Round 2

Reviewer 2 Report

Despite the organization led by the National Agency for Regional Health Services of the Ministry of Health to encourage the implementation of the Regional Oncological Networks, the recent response (2021) of the regional health services is still highly heterogeneous although we are already in the fourth national survey. Not only that, where implemented, the organizational and management reference models relating to the regional oncological networks differ from each other. This makes the existing data on the aspects connected to the trend of tumors in Italy intrinsically heterogeneous. Such an evaluation is a multidisciplinary process which must take place coherently with the other health assistance, technical-administrative, socio-economic processes of the regional health systems and of the structures that are part of them.

The evaluation criteria concern not only the health policy factors connected with the properties of the drug, its cost, which patients it is aimed at and what the state and progression of their disease is, but also which types of organizations treat the population, the training and resource burden necessary to support medical personnel in providing clinically appropriate care, interdependence, i.e. the participation of clinical services within the organization to improve the quality of medical action or to use institutional resources more efficiently. The ISPOR-SMDM model also explains that the point estimation and the uncertainty of the parametric data used for predictive models are part of a single process.

The results of an Italian cost-effectiveness analysis are therefore subject to intrinsic uncertainty, and the authors coherently showed a sensitivity analysis which highlighted how the results obtained are "sensitive" to the variation of the hypotheses or the value of some data and explained many of the weaknesses of their analysis.

I do not dispute the decision model used which is intended to provide economic forecasts, but I consider that, in any case, a model always produces an output and the output strongly depends on the positive and negative aspects of the data provided.

A model describes the probable evolution of a system, of a process, based on the initial data (initial conditions) provided by the user (the input) by returning the final data (output). So the output is a function of the input variables and the "solution", made explicit by the model, will always be delimited by the perimeter defined by the set of data provided. It is more or less the same logic of the economic forecast analyzes that are done with artificial intelligence. They cannot envisage models for analysis imposed by the big data in the reference database. All health data are Big Data for which the NHS should not only collect health data, which records an ever faster acquisition but also analyze them in real-time. The goal is to make the decision-making process as timely as possible, offering decision-makers the greatest speed and immediacy of action and reaction to social and health events that occur throughout the nation. The 5 Vs represent the fundamental principles to which any predictive analysis concerning human health based on big data must comply.

If the data, albeit optimal and qualitatively excellent, reflect only part of the variables (and not always the same) present in a limited part of the country, we will have excellent theoretical "efficacy" but poor real "effectiveness".

The authors tried to do this while minimizing the great decision-making disparities of the Italian public health, which operates completely independent of the AIFA conclusions. In the event of a lack of funds in a region (which in many regions occurs punctually around the 8th-9th month of the year) the drug, even if taken into consideration, will no longer be disbursed. The regions on which data and parameters of the economic forecasting model have been set up will have fewer problems.

I'm not discussing the result of the authors who applied a specific evaluation model, with results in line with other similar surveys, but my strong doubt is: are we really evaluating the Italian healthcare reality? Unfortunately, I have to conclude that beyond the efforts made to get and evaluate the data, the authors could not do more.